# Structural insights into the in situ assembly of clustered protocadherin γB4

Ze Zhang ®[1,2,3,8], Fabao Chen[1,8], Zihan Zhang[1,8], Luqiang Guo[4], Tingting Feng[1], Zhen Fang[1], Lihui Xin[5], Yang Yu[5], Hongyu Hu ®[2,3], Yingbin Liu[1,6,7] & Yongning He ®[1,2,3,6,7] ✉

Clustered protocadherins (cPcdhs) belong to the cadherin superfamily and play important roles in neural development. cPcdhs mediate homophilic adhesion and lead to self-avoidance and tiling by giving neurons specific identities in vertebrates. Structures and functions of cPcdhs have been studied extensively in past decades, but the mechanisms behind have not been fully understood. Here we investigate the in situ assembly of cPcdh-γB4, a member in the γ subfamily of cPcdhs, by electron tomography and find that the full length cPcdh-γB4 does not show regular organization at the adhesion interfaces. By contrast, cPcdh-γB4 lacking the intracellular domain can generate an ordered zigzag pattern between cells and the *cis*-interacting mode is different from the crystal packing of the ectodomain. We also identify the residues on the ectodomain that might be important for the zigzag pattern formation by mutagenesis. Furthermore, truncation mutants of the intracellular domain reveal different assembly patterns between cell membranes, suggesting that the intracellular domain plays a crucial role in the intermembrane organization of cPcdh-γB4. Taken together, these results suggest that both ectodomain and intracellular domain regulate the in situ assembly of cPcdh-γB4 for homophilic cell adhesion, thereby providing mechanistic insights into the functional roles of cPcdhs during neuronal wiring.

During neural development, neurons are organized into complex networks by following certain repulsive interactions, including self-avoidance and tiling, to guarantee correct arrangements and functionality of the networks[1–3]. Self-avoidance refers to the repulsion between arbors from a single neuron, during which neurons need to discriminate self from non-self[4,5]. Therefore, self-avoidance demands that a single neuron has its own specific identity distinct from thousands of others it may contact[6,7]. In tiling, different neurons with the

same functional roles would avoid each other by sharing the same identities[8,9].

In *Drosophila*, Down syndrome cell adhesion molecules 1 (DSCAM1) and DSCAM2 have been shown to play key roles in self-avoidance[10–12] and tiling[13], respectively. In vertebrates, evidence suggests that self-avoidance and tiling are mediated by clustered protocadherins (cPcdhs), which can lead to repulsion between axonal or dendritic neurites[8,14–17]. cPcdhs belong to the cadherin superfamily and

[1]State Key Laboratory of Systems Medicine for Cancer, Shanghai Cancer Institute, Renji Hospital, Shanghai Jiao Tong University School of Medicine, Shanghai, China. [2]Shanghai Institute of Biochemistry and Cell Biology, Center for Excellence in Molecular Cell Science, Chinese Academy of Sciences, Shanghai, China. [3]University of Chinese Academy of Sciences, Beijing, China. [4]Department of Molecular Biosciences, The University of Texas at Austin, Austin, TX, USA. [5]National Facility for Protein Science in Shanghai, Shanghai Advanced Research Institute, Chinese Academy of Sciences, Shanghai, China. [6]Shanghai Key Laboratory for Cancer Systems Regulation and Clinical Translation, Jiading District Central Hospital, Renji Hospital Jiading Branch, Shanghai, China. [7]Department of Biliary-Pancreatic Surgery, Renji Hospital, Shanghai Jiao Tong University School of Medicine, Shanghai, China. [8]These authors contributed equally: Ze Zhang, Fabao Chen, Zihan Zhang. ✉e-mail: heyn@shsmu.edu.cn

are named according to the clustered genomic organization[18] and general existence in distantly related species[19]. cPcdhs contain 50–60 isoforms, and the genes of cPcdhs locate on human chromosome 5[18] or mouse chromosome 18[20] and are arranged closely in three tandem clusters, which correspond to three subfamilies: α, β, and γ. Each cluster contains 10–30 variable exons, and each variable exon encodes an intact ectodomain (EC), a transmembrane domain (TM), and a variable intracellular domain (VIC). Variable exons in the γ cluster can be further divided into type A and type B[18,20]. α and γ clusters also contain three constant exons, which encode a common intracellular domain (CIC) that is conserved in all isoforms within the cPcdh subfamilies[7,21]. Therefore the intracellular domains (IC) of α and γ-cPcdhs have both VIC and CIC, while β-cPcdhs lack the cluster-specific CIC[22].

To achieve self-avoidance and tiling, cell adhesion mediated by the adhesion molecules is required for both processes[7,13,23]. Published data have shown that cells expressing the same sets of cPcdh isoforms exhibit cell adhesion, while a single isoform mismatch in the combinations would abolish adhesion[24]. Such high matching demand between repertoires means that 50–60 cPcdh isoforms can support identities for billions of neurons[25–27].

The homophilic binding of cPcdhs has been studied extensively in the past decades[24,27–30]. The ectodomains of cPcdhs contain six extracellular cadherin domains (EC1 to 6). Crystallographic results show that the *trans* homophilic interaction occurs between EC1-4 of the monomers[29–31], while EC5-6 may mediate the *cis*-dimer formation[27,29,31,32]. The alternate *trans* and *cis* interactions of cPcdh ectodomains may result in an extended zipper-like structure between membranes, as has been shown in a liposome model[33]. Imaging characterizations of the cells expressing cPcdhs have also been reported before[34], but details at the adhesion interfaces remain unclear.

In the meantime, evidence has shown that the intracellular domains of cPcdhs are involved in the activation of downstream signaling cascades[35–39], which could be important for neuronal avoidance[22,25]. Moreover, biochemical assays suggest that the intracellular domains of cPcdhs could interact with each other[40–42] and may also restrict accumulation of cPcdhs at cell-cell contacts[28,43], but the exact roles of intracellular domain in adhesion or self-avoidance have not been fully understood.

Here, we explore the in situ assembly of cPcdh-γB4 (γB4) by combining fluorescence microscopy, electron tomography (ET), and mutagenesis studies, which would provide insights for the mechanism of γB4 in mediating cell adhesion and neuronal avoidance during neural network formation.

## Results

### Full-length γB4 does not form an ordered assembly pattern at the adhesion interfaces

Crystal structure shows that the ectodomain of γB4 can generate a zipper-like pattern through alternate *trans*-interaction of EC1-4 and *cis*-interaction of EC5-6, this assembly feature was also observed in a liposome modeling system where the ectodomain of γB6 was coupled onto the liposome surfaces[33]. In order to examine the in situ assembly of γB4, we transfected HEK293 cells with the full-length mouse γB4 (γB4-FL) fused with a GFP tag at the C-terminus, and fluorescent confocal microscopy was applied to monitor the formation of cell adhesion. Images showed that green fluorescent lines were highlighted at cell-cell contacts where γB4 accumulated for adhesion (Fig. 1A and Supplementary Fig. 1A). Then the transfected cells were subjected to high-pressure freezing and freeze substitution (HPF-FS), and the plastic-embedded ultra-thin sections were prepared for electron microscopic (EM) observation[44–46]. The resulting EM images displayed some electron-dense features between the adjacent cell membranes at cell-cell contact regions (Fig. 1A and Supplementary Fig. 1B, C), which was not observed for the non-transfected cells[45]. However, no ordered assembly pattern

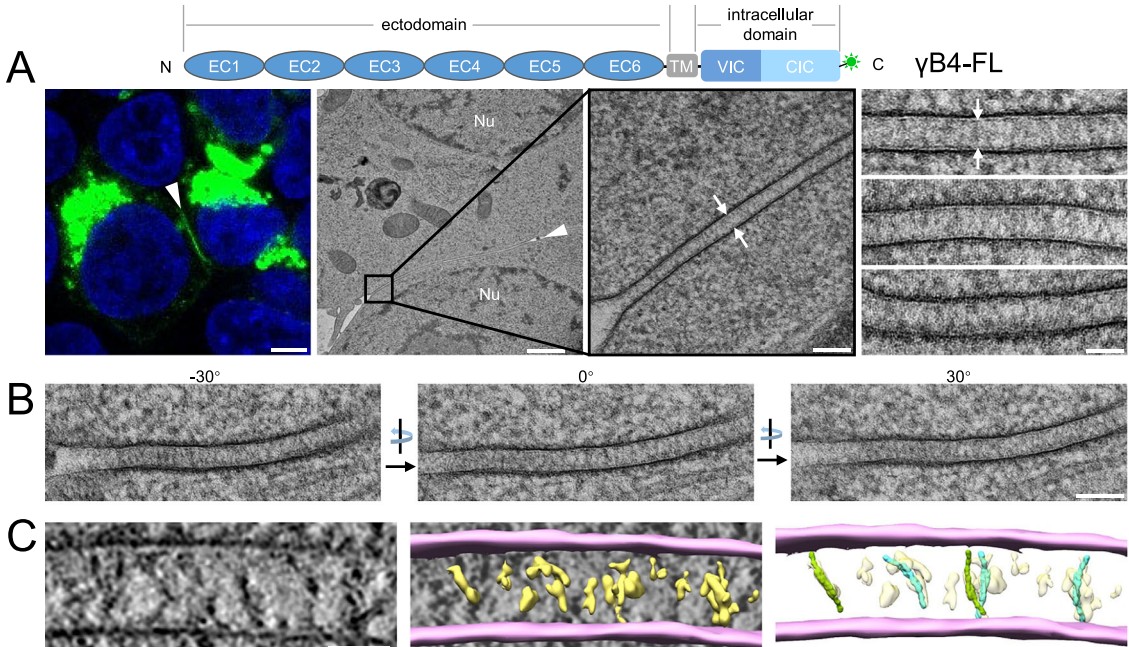

**Fig. 1 | Microscopic image of the cell adhesion interfaces by γB4-FL. A** A schematic diagram of the domain arrangement of γB4-FL is shown on the top, the GFP tag is shown in green. A confocal fluorescent image of an adhesion interface (white arrowhead) by γB4-FL is shown on the left (scale bar, 5 μm). EM images of an adhesion interface (white arrowhead) (scale bar, 1 μm) with a zoom-in view (white arrows) (scale bar, 100 nm) are shown in the middle. A gallery of the γB4-FL mediated adhesion interfaces (white arrows) is shown on the right (scale bar, 50 nm; more than ten independent interfaces are imaged). **B** EM images of a γB4-FL mediated adhesion interface visualized at different tilt angles (scale bar, 100 nm). **C** A tomographic slice of a γB4-FL mediated adhesion interface (left) (scale bar, 35 nm) and a segmentation model of the tomogram (middle). The cell membranes and the densities in between are colored pink and yellow, respectively. The densities are tentatively docked with the *trans*-dimers of the ectodomain of γB4 (green or cyan) (right).

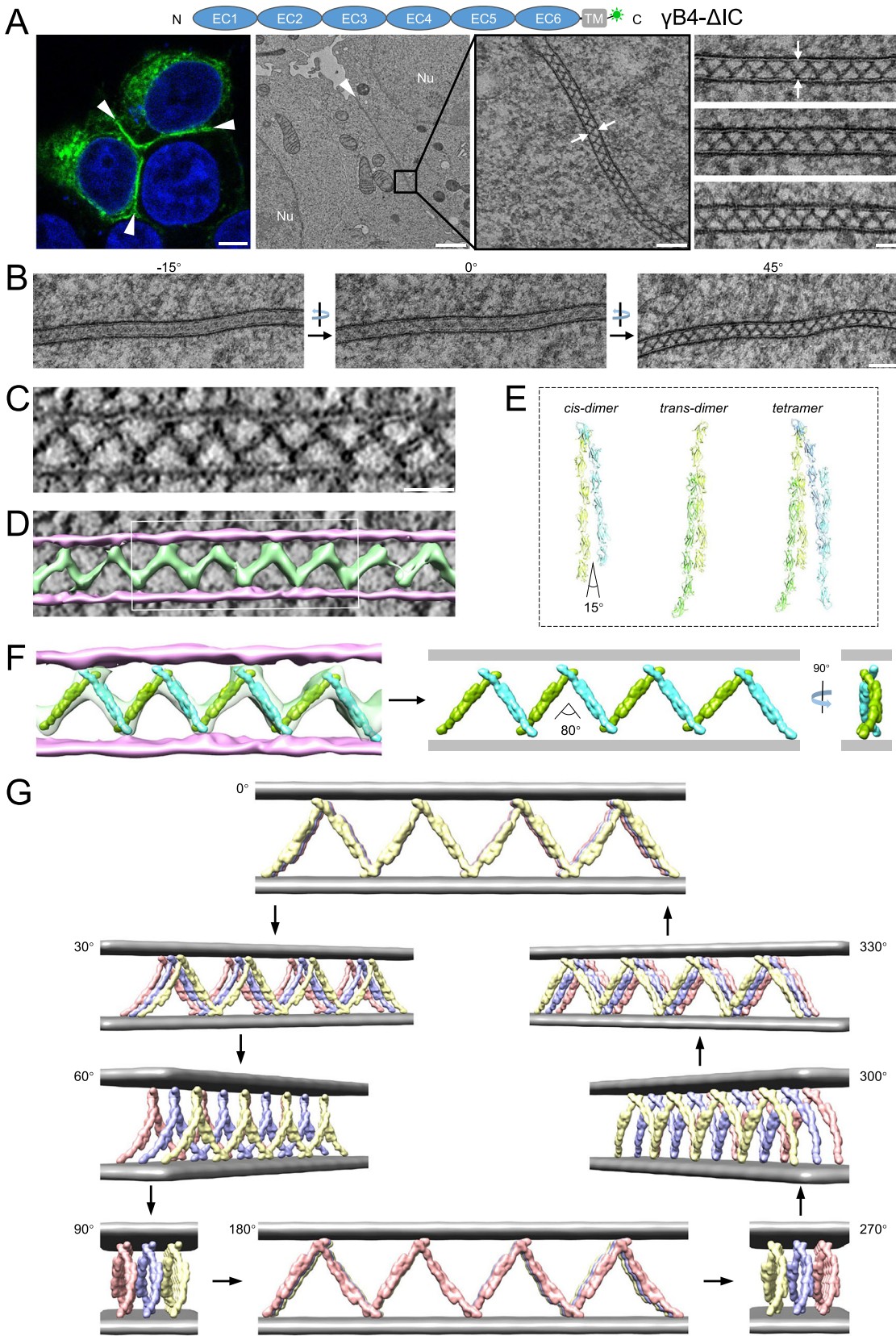

was found after inspecting a number of adhesion interfaces (more than 10 interfaces). Since the EM sections were prepared by random cuts in 3D, we also checked the interfaces with different viewing angles by rotating the specimens in the electron microscope, and no regular pattern was observed in the interfaces (Fig. 1B). Furthermore, EM tilt series were collected for tomographic reconstruction, the resulting tomograms confirmed that no ordered structure was assembled at the adhesion interfaces (Fig. 1C). After semi-automated segmentation of the tomograms[47], a few density volumes that may correspond to the *trans*-dimers of the ectodomain of γB4 could be observed and were tentatively docked by the crystal structure with poor accuracy (Fig. 1C). These data suggest that the in situ

**Fig. 2 | Microscopic images and a tomographic model of the cell adhesion interface by γB4-ΔIC. A** A schematic diagram of the domain arrangement of γB4-ΔIC is shown on the top, the GFP tag is shown in green. A confocal fluorescent image of an adhesion interface (white arrowheads) by γB4-ΔIC is shown on the left (scale bar, 5 μm). EM images of an adhesion interface (white arrowhead) (scale bar, 1 μm) with a zoom-in view (white arrows) (scale bar, 100 nm) are shown in the middle. A gallery of the γB4-ΔIC mediated adhesion interfaces (white arrows) is shown on the right (scale bar, 50 nm; more than thirty independent interfaces are imaged). **B** EM images of a γB4-ΔIC mediated adhesion interface visualized at different tilt angles (scale bar, 100 nm). **C** A tomographic slice of a γB4-ΔIC mediated adhesion interface (scale bar, 35 nm). **D** A segmentation model of the tomogram of the γB4-ΔIC mediated adhesion interface shown in (**C**). The cell membranes and the densities in between are colored pink and green, respectively. **E** The *trans-* and *cis-*dimers and tetramer of γB4 ectodomain found in crystals. The monomers are colored light green, green, cyan, or blue. **F** The segmentation model shown in (**D**, white rectangle) is fitted with the *trans-*dimers of the ectodomain of γB4 (green or cyan) (left), revealing the assembly pattern of γB4-ΔIC between cell membranes (right). **G** A 3D tomographic model of the assembly of γB4-ΔIC at the adhesion interfaces. The cell membranes are colored gray. γB4-ΔIC are shown in yellow, blue, or red.

organization of γB4-FL at the adhesion interfaces might be different from the crystal packing. In addition, the tomograms showed that the intermembrane distance of the adhesion interfaces mediated by γB4-FL was about 34 nm, rather than 38 nm according to the assembly model based on the crystal packing of γB4 ectodomain.

### γB4 lacking the intracellular domain forms an ordered zigzag pattern at the adhesion interfaces

In parallel with the experiments for γB4-FL, we also transfected HEK293 cells with the γB4 lacking the intracellular domain (γB4-ΔIC) and prepared the specimens similarly. Fluorescent confocal images confirmed that γB4-ΔIC also accumulated at the adhesion interfaces, forming highlighted green lines (Fig. 2A and Supplementary Fig. 2A). But surprisingly, EM images showed that γB4-ΔIC formed an ordered zigzag pattern between cell membranes at the adhesion interfaces (Fig. 2A and Supplementary Fig. 2B, C). We also found that the patterns could vary for different interfaces, which might be due to the different cutting angles during EM sectioning, as mentioned above (Fig. 2B, middle image). Therefore, we inspected the interfaces with different tilt angles under EM and found that the zigzag pattern could always be visualized at certain tilt angles for different interfaces (Fig. 2B), suggesting that γB4-ΔIC was stably assembled into an ordered structure at the interfaces.

To further characterize the zigzag pattern of γB4-ΔIC at the interfaces, EM tilt series were collected for tomographic reconstruction. In the tomograms, the zigzag pattern could be seen clearly (Fig. 2C), and the intermembrane distance of the adhesion interfaces mediated by γB4-ΔIC was about 28 nm, in contrast to the distance of 34 nm by γB4-FL. To build an assembly model of γB4-ΔIC, the tomograms were segmented (Fig. 2D), and the crystal structure of the ectodomain of γB4 was docked into the segmented tomograms, revealing the assembly pattern of γB4-ΔIC between cell membranes (Fig. 2E, F). During model fitting, the crystallographic *trans-*dimers of the ectodomain of γB4 matched the tomographic density reasonably well, suggesting that *trans-*dimeric interaction was maintained at the adhesion interfaces. By contrast, the *cis-*dimer in the crystals could not be fitted into the tomograms directly unless the angle between the two monomers of the *cis-*dimer increased from 15 degrees to 80 degrees (Fig. 2E, F), which would result in a reduction of the intermembrane distance to 28 nm, as observed in the tomograms (Fig. 2C).

A 3D fitting model of γB4-ΔIC was generated according to the tomogram (Fig. 2G and Supplementary Fig. 3 and Supplementary Mov. 1). In the model, the ectodomain of γB4 formed an ordered zigzag pattern between cell membranes which differs from the pattern found in crystal packing[33]. The *trans* interaction of γB4 ectodomain is retained, and arrays of γB4-ΔIC were arranged in parallel at the adhesion interfaces (Fig. 2G), which is in agreement with the serial EM sections of the interfaces (Supplementary Fig. 5) and also consistent with the tilts series with different specimen cutting angles (Supplementary Fig. 3). The transition from the crystal packing of γB4 ectodomain to the zigzag pattern between cell membranes can be achieved by increasing the angle of the *cis-*dimers like an extendable fence (Fig. 2E–G).

### EC5 is important for the zigzag pattern formation of γB4-ΔIC

The zigzag pattern formed by γB4-ΔIC between cell membranes suggests that the ectodomain of γB4 can self-assemble into the ordered structure in the membrane environment in the absence of IC. Among the EC domains of γB4, EC1-4 is involved in *trans* dimeric interaction, which is retained in the in situ assembly of γB4-ΔIC. By contrast, the *cis-*dimeric interaction mediated by EC5-6 has changed significantly, implying that they may play a major role in the zigzag pattern formation. Therefore, we made a chimeric molecule where EC5-6 of γB4 are substituted by EC5-6 of γB6 and inspected the pattern formation at the adhesion interfaces (Fig. 3A). The EM data showed that the zigzag pattern disappeared when EC5-6 of γB4 was replaced with that of γB6, confirming the importance of EC5-6 in the assembly (Fig. 3A). Then we generated two chimeric molecules, where either EC5 or EC6 of γB4 was substituted, the resulting images showed that the substitution of EC5 disrupted the zigzag pattern formation (Fig. 3B), whereas the substitution of EC6 had no impact on the pattern formation (Fig. 3C), suggesting that EC5 is crucial for the pattern formation of γB4-ΔIC.

To identify the residues on EC5 that might be important for the assembly, we did a sequence alignment of EC5 between γB4 and γB6, and the result showed a high sequence identity (87%) except in four regions: T451-V453, Q484-Y488, E497 and H535-S537 (Fig. 4A). Then we made four mutants of γB4-ΔIC by replacing the corresponding residues, including T451Q/V453S, Q484H/Y488S, E497K and H535Q/S537K (Fig. 4A). The mutants were applied for EM visualization of the adhesion interfaces, and the results showed that all the mutants except E497K, retained the zigzag pattern at the cell interfaces (Fig. 4B–E), suggesting that E497 of EC5 might play an important role in the ordered assembly of γB4-ΔIC.

### Cis-interaction of the in situ assembly of γB4-ΔIC

Tomographic model fitting suggested that the angle of the crystallographic *cis-*dimer increased from 15 degrees to 80 degrees (Fig. 5A, B). In the crystal structure, E497 is located on the surface of EC5 (Fig. 5A). Following the angle change of the *cis-*dimer, E497 might be able to approach a positively charged region on the surface of EC6 from the other monomer, which may provide electrostatic interaction to stabilize the zigzag pattern and could also explain the disruption of the zigzag pattern by the single mutation E497K (Fig. 5B).

According to the published data, residues L585 and V590 are located at the *cis-*dimeric interface of γB4 ectodomain in the crystal structure and are important for forming cell adhesion[31,32]. Here we also made mutants of the two residues, L585A and V590G, on γB4-ΔIC (Fig. 5C), and indeed, almost no adhesion interface was identified for the cells transfected with these two mutants by fluorescent microscopy (Fig. 5D, E), similar to the previous observation[31,32], implying that these residues might also be important for the assembly of γB4-ΔIC on the cell surface. Taken together, it appears that the *cis-*dimeric interface found in the crystal structure may act as a "hinge" maintained by hydrophobic interactions (Fig. 5C), while E497 may interact with the neighboring monomers through charge interaction and stabilize the large opening angle of the *cis-*dimers in the zigzag pattern of γB4-ΔIC at the adhesion interfaces (Fig. 5B).

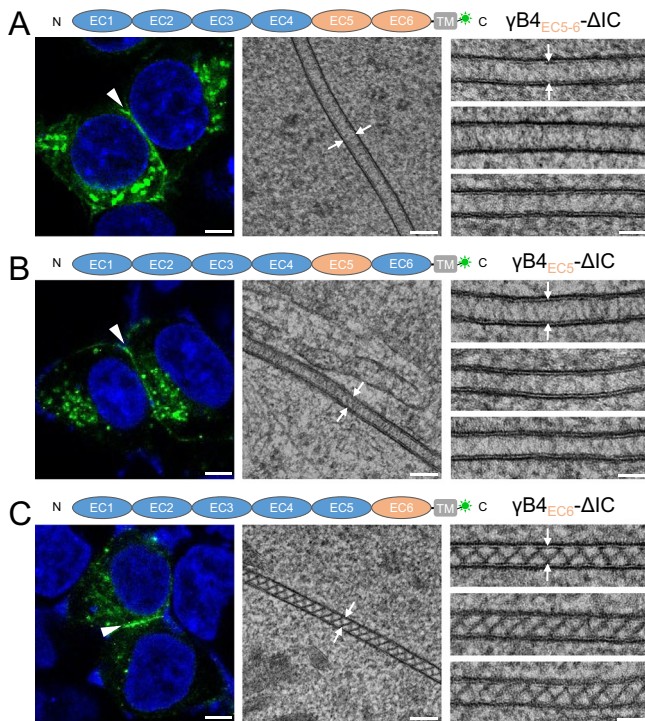

**Fig. 3 | Microscopic images of the cell adhesion interfaces by the substitutional mutants of γB4-ΔIC. A** A schematic diagram of a substitutional mutant of γB4$_{EC5-6}$-ΔIC is shown on the top. A confocal fluorescent image of an adhesion interface (white arrowhead) by the mutant is shown on the left. An EM image of an adhesion interface (white arrows) is shown in the middle. A gallery of the γB4$_{EC5-6}$-ΔIC mediated adhesion interfaces (white arrows) is shown on the right (more than fifteen independent interfaces are imaged). **B** A schematic diagram of a substitutional mutant of γB4$_{EC5}$-ΔIC is shown on the top. A confocal fluorescent image of an adhesion interface (white arrowhead) by the mutant is shown on the left. An EM image of an adhesion interface (white arrows) is shown in the middle. A gallery of the γB4$_{EC5}$-ΔIC mediated adhesion interfaces (white arrows) is shown on the right (more than twelve independent interfaces are imaged). **C** A schematic diagram of a substitutional mutant of γB4$_{EC6}$-ΔIC is shown on the top. A confocal fluorescent image of an adhesion interface (white arrowhead) by the mutant is shown on the left. An EM image of an adhesion interface (white arrows) is shown in the middle. A gallery of the γB4$_{EC6}$-ΔIC mediated adhesion interfaces (white arrows) is shown on the right (more than fourteen independent interfaces are imaged). Scale bar, 5 μm (left), 100 nm (middle), 50 nm (right).

## Intracellular domain regulates the in situ assembly of γB4

As shown above, the in situ assembly patterns of γB4-FL and γB4-ΔIC are significantly different, suggesting that the intracellular domain of γB4 is involved in regulating the organization of γB4 at the adhesion interfaces. In fact, the published data showed that the intracellular domains of cPcdhs could interact with each other[40–42], therefore may affect the assembly of the ectodomains of cPcdhs. To verify the interaction between the intracellular domains on the cell membrane, we co-transfected cells with γB4-FL (fused with RFP) and IC of γB4 (including TM and IC, fused with GFP), confocal images showed that IC could co-localize with γB4-FL at the interfaces (Fig. 6A). Furthermore, since IC of γB4 contains both VIC and CIC, we co-transfected γB4-FL (fused with RFP) with VIC or CIC (including TM and VIC or CIC, fused with GFP), and both confocal images displayed co-localization of VIC and CIC with γB4-FL (Fig. 6B, C), confirming that the intracellular domain of γB4 could interact with each other on the cell membrane during adhesion. In addition, the confocal images showed that in the absence of TM, IC alone distributed all over the cells, including the nucleus (Fig. 6D), which was consistent with the published data showing that the cleaved fragments of the intracellular domains of

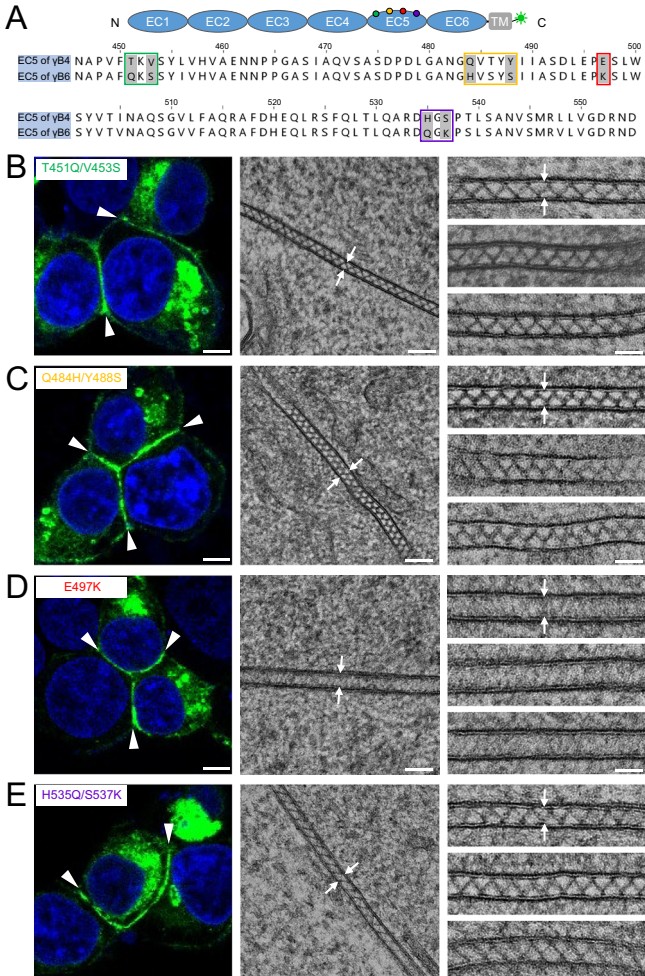

**Fig. 4 | Microscopic images of the cell adhesion interfaces by the EC5 mutants of γB4-ΔIC. A** A schematic diagram of γB4-ΔIC is shown on the top. The sequence alignment of EC5 from γB4 and γB6 is shown at the bottom. The residue differences between γB4 and γB6 on EC5 are labeled in green, yellow, red, or purple. **B** A confocal fluorescent image of an adhesion interface (white arrowheads) by the mutant T451Q/V453S is shown on the left. An EM image of an adhesion interface (white arrows) is shown in the middle. A gallery of the mutant-mediated adhesion interfaces (white arrows) is shown on the right (more than ten independent interfaces are imaged). **C** A confocal fluorescent image of an adhesion interface (white arrowheads) by the mutant Q484H/Y488S is shown on the left. An EM image of an adhesion interface (white arrows) is shown in the middle. A gallery of the mutant-mediated adhesion interfaces (white arrows) is shown on the right (more than eight independent interfaces are imaged). **D** A confocal fluorescent image of an adhesion interface (white arrowheads) by the mutant E497K is shown on the left. An EM image of an adhesion interface (white arrows) is shown in the middle. A gallery of the mutant-mediated adhesion interfaces (white arrows) is shown on the right (more than thirteen independent interfaces are imaged). **E** A confocal fluorescent image of an adhesion interface (white arrowheads) by the mutant H535Q/S537K is shown on the left. An EM image of an adhesion interface (white arrows) is shown in the middle. A gallery of the mutant-mediated adhesion interfaces (white arrows) is shown on the right (more than five independent interfaces are imaged). Scale bar, 5 μm (left), 100 nm (middle), 50 nm (right).

cPcdhs had nuclear localization[48–50]. These results suggest that the *cis*-interaction between the intracellular domains may not be very strong and only occurs locally when they stay on the cell membrane.

To further explore the impact of the intracellular domain on the in situ assembly of γB4, we generated two IC-truncation mutants, γB4-ΔCIC and γB4-ΔVIC, where CIC or VIC was removed from IC. Fluorescent microscopy showed that both mutants could mediate the

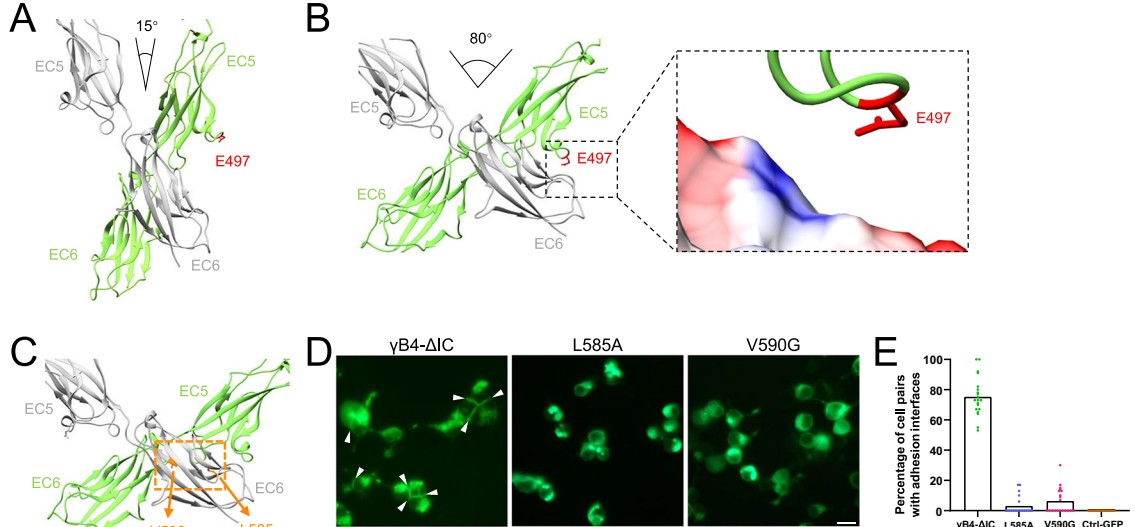

**Fig. 5 | *Cis*-dimeric interaction of the in situ assembly of γB4-ΔIC. A** *Cis*-dimeric interaction of γB4 ectodomain in the crystals. EC5 and EC6 from the two monomers are colored gray or green. The position of E497 from one of the monomers is labeled. **B** *Cis*-dimeric interaction of γB4-ΔIC on the cell surface. EC5 and EC6 from the two monomers are colored gray or green. The position of E497 from one of the monomers is labeled. A positively charged region (blue) from EC6 that may be approached by E497 during the in situ assembly of γB4-ΔIC is also shown (dashed rectangles). **C** The potential *cis*-dimeric interface of γB4-ΔIC on the cell surface (dashed orange rectangle). The positions of L585 and V590 are labeled. **D** The fluorescent images of cell adhesion mediated by γB4-ΔIC and γB4 mutants (L585A and V590G). The adhesion interfaces are indicated by white arrowheads (scale bar, 15 μm). **E** The statistics of the adhesion interfaces by γB4-ΔIC and γB4 mutants (L585A and V590G). Each dot represents the percentage of highlighted fluorescent interfaces that appeared in the pairs of neighboring cells in a stochastic field of view. A total of twenty dots were collected for each construct (five views per experiment and repeated four times). The means of the data are plotted and also provided as a source data file. The GFP-transfected cells are applied as a control.

adhesion of the transfected cells (Fig. 7A, C). EM images and the tomograms showed that the two mutants did not form ordered assemblies between cell membranes (Fig. 7A–D), and the intermembrane distances of the two mutants determined by EM were about 33 nm, similar to that of γB4-FL (Fig. 7E). However, the intermembrane tomographic densities of the two mutants were different, γB4-ΔVIC appeared to have more molecules at the interface than γB4-ΔCIC (Fig. 7B, D), and both γB4-ΔVIC and γB4-ΔCIC showed more molecules at the interfaces than γB4-FL (Figs. 1C and 7B, D). This is in agreement with the previous data showing that the intracellular domains may restrict the accumulation of cPcdhs at cell-cell contacts[28,43], and suggests that partial deletion of the intracellular domain may reduce its impact on the organization of the ectodomain, and VIC may have larger impacts on the assembly than CIC, which is not surprising as VIC locates closer to the cell membrane than CIC. In addition, we also evaluated the efficiency of adhesion formation mediated by γB4-FL and the IC-truncation mutants. The resulting statistics showed that γB4-ΔIC had the highest adhesion efficiency, implying that the zigzag pattern was preferred for the ectodomain, and the intracellular domain reduced adhesion efficiency significantly (Fig. 7F). Deletion of VIC or CIC increased the adhesion formation and may also partially recover the assembly of the ectodomain at the interfaces (Fig. 7F and Supplementary Fig. 4).

To further explore the functional region of VIC, we divided the VIC of γB4 into three regions, VIC1, VIC2, and VIC3 (Supplementary Fig. 6A), and the deletion mutants of each of the regions were applied for adhesion assays. The fluorescent images showed that among the three deletion mutants, γB4-ΔVIC2 exhibited the highest efficiency in forming adhesion interfaces (Supplementary Fig. 6B, C), suggesting that the residues in VIC2 might be important for regulating assembly formation, which is consistent with the previous data showing that the residues in this region are involved in mediating interactions of intracellular domains and trafficking[40,42,51]. In addition, structural prediction by AlphaFold[52,53] shows that the intracellular domains of cPcdhs are rather flexible without secondary structure, how they regulate the

molecular organization in situ still need further investigation in the future.

## Discussion

Cell adhesion molecules such as cadherins and IgSF adhesion molecules are important for mediating cell-cell contacts. The structures of these molecules, especially the ectodomains, have been studied extensively by X-ray crystallography in the past decades[25]. Recent developments in EM provide opportunities to visualize their in situ organizations on the cell surface[46,54], which advances the understanding of the mechanisms of cell adhesion. The in situ studies of IgSF adhesion molecules showed that their assemblies are mainly regulated by the ectodomains through *trans* and *cis* as well as membrane interactions, and some molecules may form ordered assemblies at the interfaces[45,55]. It has been proposed that the ordered assembly of the ectodomains may affect the downstream signaling or cytoskeletal organization[25,56]. However, whether an ordered organization is a general feature for the assemblies of adhesion molecules still needs further investigation.

The crystal structure of the ectodomain of γB4 shows that EC1-4 mediate *trans* dimer formation[29–31] and EC5-6 are responsible for forming asymmetric *cis*-dimers, where EC5-6 of a monomer interacts with EC6 of the other monomer[29,32]. The alternate *trans* and *cis*-dimerizations can produce a zipper-like assembly in crystal packing (Fig. 8A), and a similar pattern has been observed in a liposome model for γB6[33]. However, the in situ imaging of the adhesion interfaces by γB4-FL does not show an ordered assembly pattern, suggesting that in situ assembly could be more complex and some interactions might be missing in crystal packing. Indeed, when the intracellular domain of γB4 is removed, its intermembrane organization changes dramatically by forming an ordered zigzag pattern. The tomographic model shows that the zigzag pattern can be generated by the *trans* and *cis* dimers of the ectodomain of γB4, where the *trans*-dimer is similar to that in the crystals, but the *cis*-dimer adopts a larger opening angle between the monomers, suggesting that the ectodomain of γB4 can self-assemble

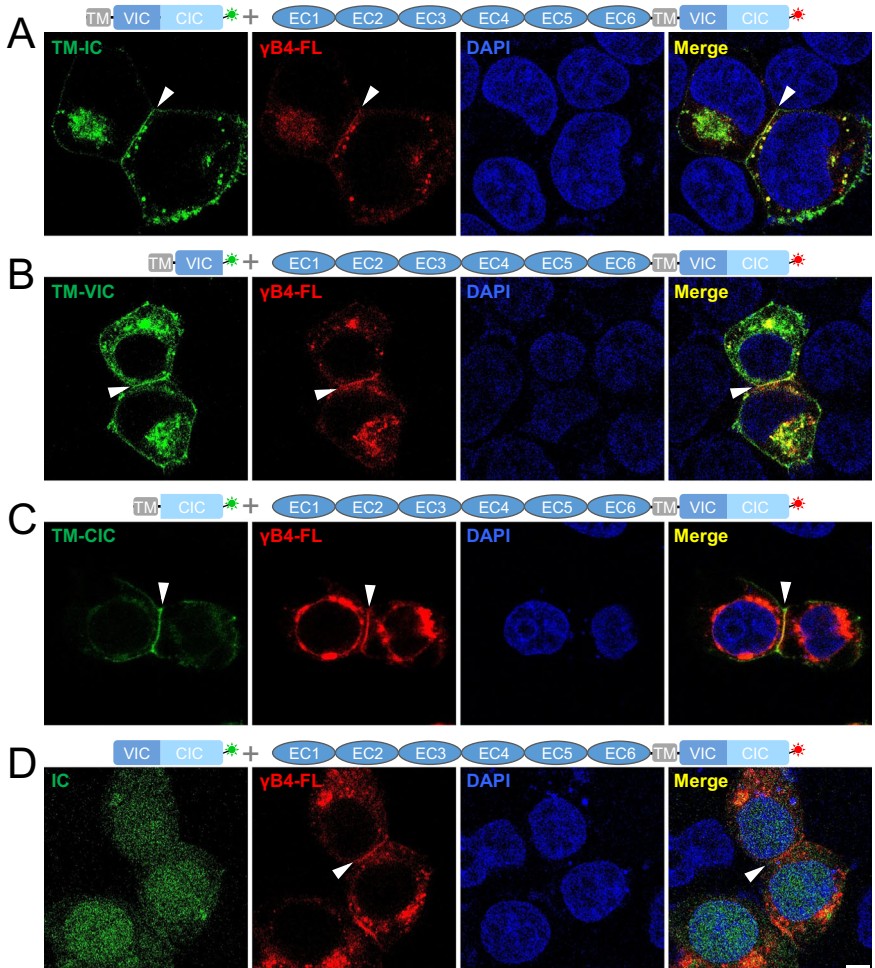

**Fig. 6 | Confocal fluorescent images of the cells co-transfected with γB4-FL and the IC mutants of γB4. A** Confocal image of the cells co-transfected with γB4-FL (red) and TM-IC (green). **B** Confocal images of the cells co-transfected with γB4-FL (red) and TM-VIC (green). **C** Confocal images of the cells co-transfected with γB4-FL (red) and TM-CIC (green). **D** Confocal images of the cells co-transfected with γB4-FL (red) and IC (green). DAPI is applied to stain the nucleus. The adhesion interfaces are indicated by white arrowheads. More than ten independent interfaces are imaged for each specimen. Scale bar, 5 μm.

into ordered structures on the cell surface, which might be driven by the forces not present in crystals. The subsequent mutagenesis data show that the *cis*-dimeric interface formed between EC5-6 and EC6 is probably or at least partially retained, and E497 on EC5 may facilitate the opening of the *cis*-dimers through charge interaction and stabilize the zigzag pattern. In addition, the sequence alignment shows that E497 only occurs in γB4 rather than other cPcdh-γ members (Supplementary Fig. 6A), implying that these members may form different assembly patterns, which needs to be clarified in the future.

The different assembly patterns of γB4-FL and γB4-ΔIC suggest that the intracellular domain also regulates the organization of γB4 between cell membranes, which may not be entirely unexpected as previous data have shown that the intracellular domains of cPcdhs can interact with each other[40,41] and is also in agreement with our fluorescent observation. Moreover, tomographic data suggest that both VIC and CIC could affect the assembly of γB4 on the cell surface, probably mainly on the *cis* organization of the molecules, and VIC seems to have larger impacts on the assembly due to its proximal location to the membrane. Based on this information, the schematic models for the in situ assembly of γB4-ΔIC and γB4-FL are generated (Fig. 8B, C). In addition, the intermembrane distance for γB4 appears to be similar to that of desmosomes[57], but the *trans* and *cis* interacting modes of cPcdhs are different from the cadherins in desmosomes, which might correspond to their different functional roles in mediating cell adhesion.

Published data have shown that *cis*-interactions between the ectodomains of cPcdhs are important for homophilic combinatorial cell recognition in mediating self-avoidance[27,33,58,59]. Here we show that the intracellular domain of γB4 also regulates the inter-membrane assembly pattern, especially the *cis* organization of the molecule, suggesting that it might be crucial in establishing homophilic adhesion between cells. Although the in situ assembly model of γB4-FL represents a simple case with only γB4 on the cell surface (Fig. 8C), it may provide clues for the situation where combinatorial expression of cPcdhs occurs on the membrane. In fact, the sequences of the intracellular domains of cPcdhs are rather diverse, all isoforms have VIC, which contains variable sequences, and α- and γ-cPcdhs have cluster-specific CIC, which is missing in β-cPcdhs[18,20]. Evidence has also shown that the intracellular domains of different cPcdhs could interact with each other[40,41]. Therefore, it might be possible that the *cis*-interactions among the isoforms may help to generate complex but specific assembly patterns like 2D barcodes on the cell surfaces for each set of cPcdhs and lead to homophilic adhesion between cells, and a single isoform mismatch would result in a different assembly pattern and disrupt the adhesion for self-avoidance. Overall, these data suggest that both ectodomains and intracellular domains of cPcdhs contribute to the homophilic adhesion between cells, but their exact roles and mechanisms still need further investigation in the future.

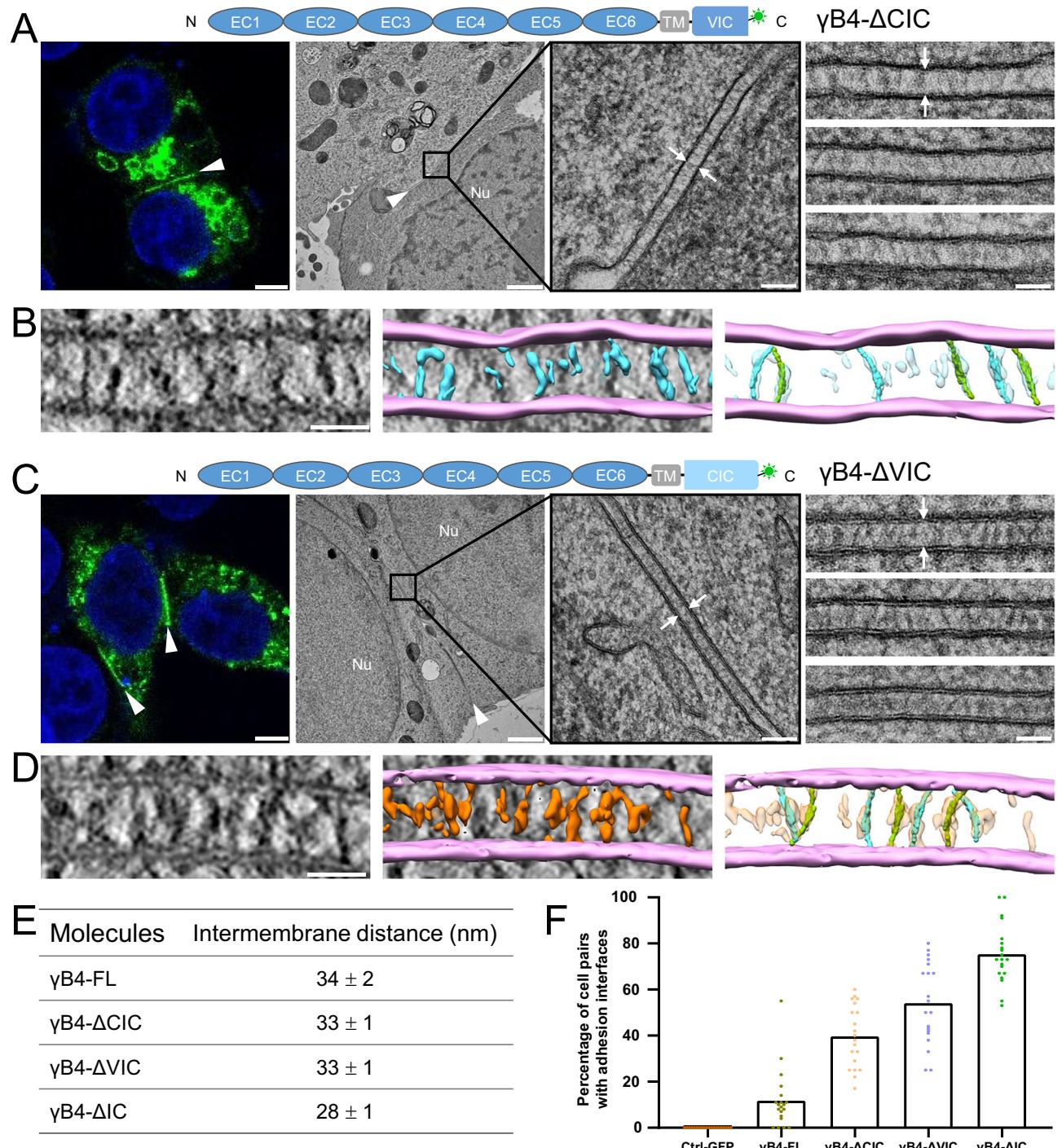

**Fig. 7 | Microscopic images and the statistics of the adhesion interfaces by the IC-truncation mutants of γB4. A** A schematic diagram of γB4-ΔCIC is shown on the top. A confocal fluorescent image of an adhesion interface (white arrowhead) by γB4-ΔCIC is shown on the left (scale bar, 5 μm). EM images of an adhesion interface (white arrowhead) (scale bar, 1 μm) with a zoom-in view (white arrows) (scale bar, 100 nm) are shown in the middle. A gallery of the γB4-ΔCIC mediated adhesion interfaces (white arrows) is shown on the right (scale bar, 50 nm; more than six independent interfaces are imaged). **B** A tomographic slice of a γB4-ΔCIC mediated adhesion interface (left) (scale bar, 35 nm) and a segmentation model of the tomogram (middle). The cell membranes and the densities in between are colored pink and cyan, respectively. The densities are tentatively docked with the *trans*-dimers of the ectodomain of γB4 (green or cyan) (right). **C** A schematic diagram of γB4-ΔVIC is shown on the top. A confocal fluorescent image of an adhesion interface (white arrowhead) by γB4-ΔVIC is shown on the left (scale bar, 5 μm). EM images of an adhesion interface (white arrowhead) (scale bar, 1 μm) with a zoom-in

view (white arrows) (scale bar, 100 nm) are shown in the middle. A gallery of the γB4-ΔVIC mediated adhesion interfaces (white arrows) is shown on the right (scale bar, 50 nm; more than eleven independent interfaces are imaged). **D** A tomographic slice of a γB4-ΔVIC mediated adhesion interface (left) (scale bar, 35 nm) and a segmentation model of the tomogram (middle). The cell membranes and the densities in between are colored pink and orange, respectively. The densities are tentatively docked with the *trans*-dimers of the ectodomain of γB4 (green or cyan) (right). **E** Statistics of the intermembrane distances of the in situ assemblies of γB4-FL, γB4-ΔCIC, γB4-ΔVIC, and γB4-ΔIC. **F** The statistics of the adhesion interfaces by γB4-FL, γB4-ΔCIC, γB4-ΔVIC, and γB4-ΔIC. Each dot represents the percentage of highlighted fluorescent interfaces that appeared in the pairs of neighboring cells in a stochastic field of view. A total of twenty dots were collected for each construct (five views per experiment and repeated four times). The means of the data are plotted and also provided as a source data file. The GFP-transfected cells are applied as a control. The data for γB4-ΔIC and GFP are also shown in Fig. 5E.

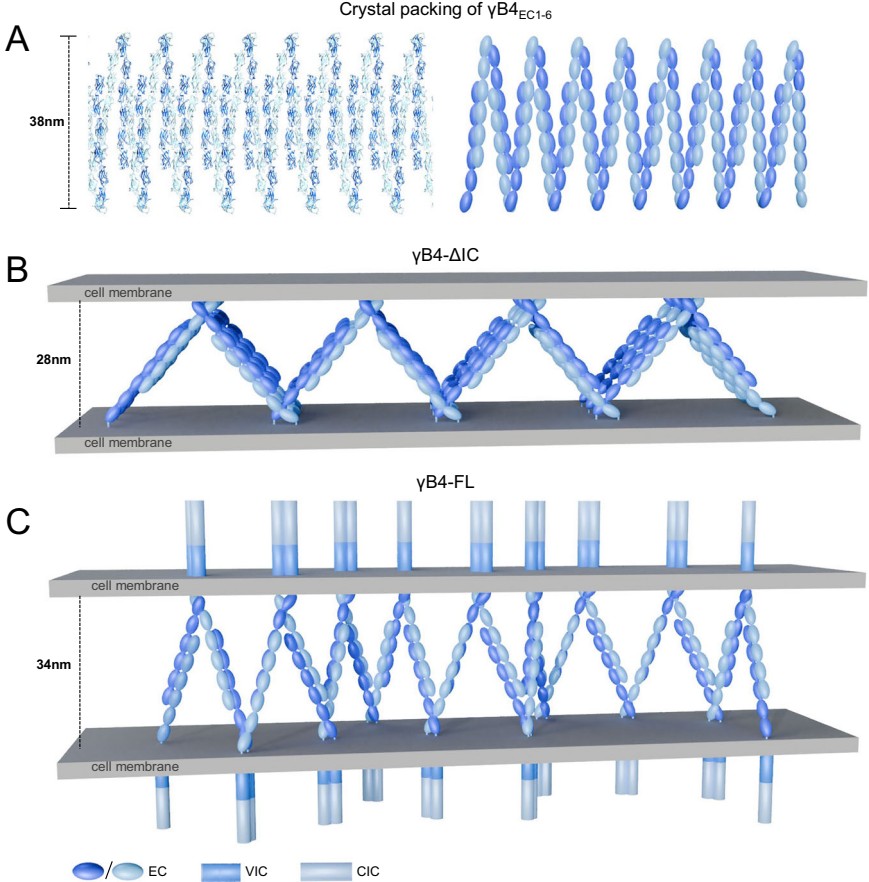

**Fig. 8 | Crystal packing of the ectodomain of γB4 and the in situ assembly models for γB4-ΔIC and γB4-FL. A** Crystal packing of the ectodomain of γB4 (left). A schematic model is shown on the right. **B** An in situ assembly model of γB4-ΔIC. **C** An in situ assembly model of γB4-FL.

## Methods

### Preparation of DNA constructs
cDNA of the full-length mouse cPcdh-γB4 (GenBank: AAI38702.1, residues 1-912, residue number includes the signal peptide) was cloned into pCMV expression vector fused with a GFP (enhanced GFP, Gen-Bank: AAB02572) or mCherry (GenBank: AAV52164) tag at the C-terminus for γB4-FL. The truncation mutants γB4-ΔIC (residues 1-720), γB4-ΔVIC (residues 1-720, 789-912), γB4-ΔCIC (residues 1-788), γB4-ΔVIC1 (residues 1-720, 741-912), γB4-ΔVIC2 (residues 1-740, 764-912), γB4-ΔVIC3 (residues 1-763, 789-912), IC (residues 721-912) and TM-IC (residues 1-30, 663-912) were generated by deletion on γB4-FL. TM-VIC (residues 1-30, 663-788) and TM-CIC (residues 1-30, 663-720, 789-912) were generated by deletion on TM-IC. For the domain substitution mutants, EC5(residues 446-555) and/or EC6 (residues 556-662) of γB4 were replaced by the EC5 (residues 448-557) and/or EC6 (residues 558-664) of γB6 by homologous recombination using the *ClonExpress MultiS One Step Cloning kit* (Vazyme, C113-01). The single or double mutants, including T451Q/V453S, Q484H/Y488S, E497K, H535Q/S537K, L585A and V590G were generated by site-directed mutagenesis on γB4-ΔIC. The primers are shown in Supplementary Table 1. All the constructs were validated by DNA sequencing.

### Confocal microscopy
HEK293T cells (NCACC, Serial: GNHu17) were cultured on coverslips coated with Poly-L-lysine (Sigma, P4707-50ML). Plasmid constructs were transfected into the cells by using Lipofectamine 2000 reagent (Invitrogen, 11668019). After 24 h, transfected cells were fixed with 4% paraformaldehyde, permeabilized with 0.5% Triton-X and mounted with antifade mounting medium with DAPI (Beyotime, P0131-25ml). Images were acquired on a confocal microscope Leica TCS SP8. For the

adhesion formation statistics, cells were cultured on multiple well plates and transfected with the γB4 constructs. After 24 h, cells were visualized under a fluorescence microscope ECLIPSE Ts2, the percentage of highlighted fluorescent interfaces appeared in the pairs of neighboring cells were counted. The experiments were repeated more than three times (five views each time). The data were analyzed by GraphPad Prism 9.0.

### EM sample preparation
Sapphire discs were marked by carbon evaporation and coated with poly-L-lysine for cell culture. HEK293T cells were transfected with the target constructs. After 24 h, the sapphire discs were transferred to specimen holders and covered by aluminum planchettes with 25-μm inner depth and filled with hexadecane. The specimens were then loaded onto a Wohlwend HPF Compact 2 high-pressure freezer (M.Wohlwend GmbH) for HPF. Frozen specimens were transferred into cryotubes containing 0.1% uranyl acetate, 0.6% water, and 1% osmium tetroxide in acetone at liquid nitrogen temperature. Freeze substitutions were completed as previously described[45,46,55], cells were then embedded into resin blocks and solidified. The resin blocks were subjected to thin sectioning on a Leica EM UC7 ultra-microtome. Ultra-thin sections of 100 nm thickness were collected onto formvar-coated copper grids with an evaporated carbon film and stained with 3% uranyl acetate at 4°C for 7 min, then by lead citrate at room temperature for 3 min. The stained sections were loaded onto a 120 kV Tecnai T12 microscope (Thermo Fisher Scientific) for imaging.

### Electron tomography
Ultra-thin sections were loaded onto a FEI Tecnai G2 electron microscope (120 kV) for collecting tomographic tilt series. Single-axis

tilt series were collected ranging from −60° to 60° with 1.5° increments at a pixel size of 1.71 Å using Xplore3D (FEI). Tomograms were reconstructed using EMAN2.9, and the final tomograms were binned with a resulting pixel size of 6.86 Å[60,61]. Fiji[62] was applied for measuring the intermembrane distances and calculating the histograms of the intermembrane densities in the tomograms. Segmentation was done semi-automatically by EMAN2.9 combined with IMOD[47,63].

## Model building
The *trans*-dimers of γB4 in the crystal structure (PDB entry 6E6B)[33] were fitted into the segmented tomographic volumes by UCSF Chimera[64,65]. The correlation coefficients were 0.71, 0.79, 0.75, and 0.76 for γB4-FL, γB4-ΔIC, γB4-ΔCIC and γB4-ΔVIC, respectively. The movie was also made by UCSF Chimera. The schematic models of γB4-ΔIC and γB4-FL were built using Blender (https://www.blender.org).

## Reporting summary
Further information on research design is available in the Nature Portfolio Reporting Summary linked to this article.

## Data availability
Source data are provided as a source data file. Crystal structure of the ectodomain of γB4: 6E6B. Source data are provided in this paper.

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

## Acknowledgements

We thank the Electron Microscopy and Integrated Laser Microscopy Systems at the National Facility for Protein Science in Shanghai (NFPS), Shanghai Advanced Research Institute, Chinese Academy of Sciences, China, for technical support. This work is supported by the National Natural Science Foundation of China (No. 32241022 and 32271243) to Y.H., and we also thank the support from the Innovative research team of high-level local universities in Shanghai (SHSMU-ZLCX20212601) to Y.H.

## Author contributions

Ze Z. contributed to the investigation, methodology, validation, and writing. F.C., Zihan Z., L.G., T.F., Z.F., L.X., Y.Y., H.H., and Y.L. contributed to the investigation and methodology. Y.H. contributed to supervision, methodology, resources, funding acquisition, and writing.

## Competing interests

The authors declare no competing interests.
