## [Transparent Peer Review file · Nature Communications]

Structural insights into the in situ assembly of clustered protocadherin γ B4

Corresponding Author: Professor Yongning He

Version 0:

Reviewer comments:

Reviewer #1

(Remarks to the Author)

Clustered Protocadherin (Pcdh) proteins confer neurons with their cell-surface identity required for neural self-avoidance. Previous biochemical and biophysical studies performed in vitro suggested a model by which Pcdh-mediated self-avoidance necessitates the assembly of a zipper-like extracellular structure whereby the extracellular domains of Pcdh proteins from membranes carrying the same Pcdh isoforms oligomerize in trans.

In this manuscript, Zhang, Chen, Zhang et al provide the first investigation of these assemblies in cells. By performing their study in HEK293 cells focusing on PcdhyB4 extracellular domain and full length protein, their study support previous published in vitro biochemical and biophysical data as well as uncovered new insight on the role of the intracellular domain (ICD) of Pcdhy proteins in the regulation of these structures.

Specifically, the authors found that the presence of the ICD changes the nature of the structure and renders the formation of the zipper-like lattice more disordered. As the ICD has been suggested to drive neural self-repulsion, these studies provide an initial molecular clue by which the ICD could function in vivo.

The authors should consider addressing the following comments to improve the logic of the manuscript:

1. Regarding the EC5 chimera experiment – when EC5-6 (and eventually, EC5 and its pertinent residue E497) of γ B4 are substituted with the residues of γ B6, the ordered zigzag pattern is lost. Does that mean that γ B6 Δ IC does not form an ordered zipper at adhesion interfaces? If so, is the zipper assembly behavior of γ B4 a unique feature of that isoform, or can this behavior be generalized to other Pcdhy isoforms? Or, on the contrary, is γ B6 different from all other Pcdhy isoforms? A sequence alignment across all Pcdhy genes (perhaps testing the conservation of E497) would help contextualize this important observation. Further, given that Pcdh undergo cis-heterodimerization (Goodman 2022, Thu 2014), can the authors speculate how isoform-specific assembly behaviors impact the organization of lattice comprised of several isoforms?
2. Figure 6 would benefit from the inclusion of schematics illustrating the differences between the transfected isoforms.
3. The authors present data that the VIC and CIC have varying effects on adhesion assemblies, namely, that the VIC exerts a larger impact on the assembly structure. As VIC-dependent trafficking has been observed before in other isoforms (such as PcdhyA3) we wonder whether there may be conserved residues within the VIC that may regulate ectodomain assembly across the Pcdhy subcluster?
4. The fluorescence colocalization data presented in Figure 6 would be more compelling if accompanied by immunoprecipitation experiments.
5. We found lines 250-254 to be confusing and therefore suggest that the authors rewrite this sentence to improve clarity. Further, regarding this data, could the authors quantify the levels of intermembrane densities as Figure S4 is confusing? Maybe the authors can integrate the area under the curve to quantify densities?
6. I believe that the Bonn. S et paper cited is the one published in 2007, not 2023.

Reviewer #2

(Remarks to the Author)

In the presented manuscript, Zhang, Chen, and Zhang et al. study the structure of the gamma-B4 clustered protocadherin by electron microscopy of plastic-embedded HEK293 cells and complement these studies with immunofluorescence microscopy and mutagenesis studies. By electron tomography, the authors show that the gamma-B4 deletion mutant of the intracellular domain forms ordered clusters in transfected HEK293 cells. These structures are absent when transfecting the full length gamma-B4 constructs. By mutagenesis and subsequent transmission electron microscopy, the authors show that the EC5 domain of gamma-B4 is driving the cis-interactions within the clustered arrays and swapping the domain with the EC5 domain of another protocadherins (gamma-B6) disturbs the interaction and, consequently, the formation of clustered arrays. They demonstrate that a number of point mutation within the EC5 domain, E497K, L585A, and V590G, interferes with array formation, identifying the likely interaction interface of gamma-B4 that leads to cluster formation in absence of the IC domain.

I think it is the presented work is interesting regarding the mediation and regulation of cell-cell interactions. As my expertise is not in the biology of Protocadherins, I will focus myself on the technical aspects of the manuscript while adding comments on the biology and modeling of the observed data wherever the text remains unclear.

Major comments

1. A major concern to me is the model derived in Fig. 2, especially regarding the arrangement of the array in Z. How do the authors justify the incorporation of multiple copies in Z? When going into the cross section of the tomogram, one can see that the pattern seems to remain the same as in the direction shown in the Figure 2C rather than an array of structures as shown in Figure 2G 90° and 270°. See an image added for clarification (Image.png in attachment). The current data presented is from ~50-70 nm sections. It would be helpful to perform tomography experiments from thicker sections to obtain more information in the Z direction to clarify the 3D organization of the clustered arrays.
2. The authors state that l. 146 - l. 148 "... we also found that the patterns could vary for different interfaces, which might be due to the different cutting angles during EM sectioning as mentioned above." How does this fit to their model? Maybe I missed them, but showing tomograms of these different angles estimating the cutting angle and discussing this more clearly could improve the justification of their model.
3. It is unclear to me whether the preparations are done correlatively. If so, it would be helpful to show overlays of the clustered arrays with the GFP signal in the supplementary information. If the experiments were not done by CLEM, the authors should comment on how they are certain that the clustered regions are not simply outside the imaged sections and may still be found in earlier or later sections of the same cell.
4. What do the authors suggest is the mechanism of the inhibition of array formation of the intracellular domain of gamma-B4? While it may be beyond the scope of this study, discussing potential mechanisms would be interesting.
5. Which GFP construct was used? Is the construct prone to dimerization and could this have an effect on the observed clustering? Is the effect also observed in non-GFP tagged constructs?
6. Figure 1 C, Figure 2 G: docking crystal structures into these densities without proper sampling methods is not very helpful. At the minimum, systematic fitting (template matching) should be performed. More rigorous sampling methods would be helpful to get statistics (e.g. with the integrative modeling platform (IMP), Sali Lab) on these assemblies.
7. Regarding the Reporting summary, there is a number of things that need attention. First, in the replication statement, it says "Experiments are reproducible". This is not very informative and should have the number of experiments run per condition.
8. No "Data availability" statement is given in the manuscript.

Minor comments

1. Figure 1: indicate cytosol and extracellular space of the domain schematic would make the orientation of the domains clearer to the reader.
2. l.136-137: Figures should be cited in order as it breaks the reading flow, it would be worthwhile to consider rephrasing.
3. cPcdh abbreviation introduced in the abstract but not in the introduction
4. Why does the extracellular space get stained by DAPI in the images of Figure 6?
5. l. 271-273: Sentence structure should be revised.
6. l. 299-300: These forces (or interactions) are not "not identified" in crystals, but simply not present.

7. Out of curiosity, how does the structure of cPcdhs compare to the desmosomes studied by the Frangakis group, Sikora et al. <https://www.pnas.org/doi/epdf/10.1073/pnas.2004563117> ? The membrane-to-membrane distance and proposed models seem to be at least similar for the full-length gamma-GB4, thus citing and comparing to these studies may be adequate.

Reviewer #3

(Remarks to the Author)

This paper represents an important contribution from the He group. Clustered protocadherin ectodomains have been shown to have two dimer interfaces and, together in vitro, they enable the assembly of a zig-zag-like zipper pattern. Whether this zipper pattern exists in vivo is a key question for protocadherin biology. Here, He and collaborators, express clustered protocadherin B4 on the surface of HEK 293 cells, and investigate the structures they form by cryo-EM tomography. Protocadherin B4 was previously used in crystallographic and liposome-based tomographic studies.

When full-length protocadherin B4 protein – containing the cytoplasmic domain – is expressed on the cell surface, and the junctions between apposed cells are examined, no regular zig-zag pattern is observed. The investigators then produced a construct lacking the cytoplasmic domain, and remarkably a zig-zag pattern appears, though it is somewhat different from the crystal-packing lattice previously observed between liposomes with His-tagged protocadherin, having a more obtuse cis-dimer angle. The authors also show that replacement of EC5-EC6, which forms the cis protocadherin dimer interface, leads to the lack of zipper formation.

Based on these observations, the authors propose the quite reasonable suggestion that the cytoplasmic domain can play a role in the structure and assembly of the extracellular region.

The paper is very clearly written, and I have no major scientific issues with the results or their presentation.

Minor issues:

The authors say that EC5-EC6 was “replaced”. Replaced with what?

Reviewer #4

(Remarks to the Author)

Version 1:

Reviewer comments:

Reviewer #1

(Remarks to the Author)

The authors have addressed our comments. We therefore recommend this manuscript for publications. Congratulations on an important finding for the field.

Reviewer #2

(Remarks to the Author)

The authors have addressed all my previous comments. The serial section data presented in Figure S5 support the consistency with the suggested model for protocadherin zigzag clustering. I congratulate them on the exciting paper.

Reviewer #4

(Remarks to the Author)

Response letter

Reviewer comments

Reviewer #1 (Remarks to the Author):

Clustered Protocadherin (Pcdh) proteins confer neurons with their cell-surface identity required for neural self-avoidance. Previous biochemical and biophysical studies performed in vitro suggested a model by which Pcdh-mediated self-avoidance necessitates the assembly of a zipper-like extracellular structure whereby the extracellular domains of Pcdh proteins from membranes carrying the same Pcdh isoforms oligomerize in trans.

In this manuscript, Zhang, Chen, Zhang et al provide the first investigation of these assemblies in cells. By performing their study in HEK293 cells focusing on Pcdh γ B4 extracellular domain and full length protein, their study support previous published in vitro biochemical and biophysical data as well as uncovered new insight on the role of the intracellular domain (ICD) of Pcdh γ proteins in the regulation of these structures.

Specifically, the authors found that the presence of the ICD changes the nature of the structure and renders the formation of the zipper-like lattice more disordered. As the ICD has been suggested to drive neural self-repulsion, these studies provide an initial molecular clue by which the ICD could function in vivo.

The authors should consider addressing the following comments to improve the logic of the manuscript:

1. Regarding the EC5 chimera experiment – when EC5-6 (and eventually, EC5 and its pertinent residue E497) of γ B4 are substituted with the residues of γ B6, the ordered zigzag pattern is lost. Does that mean that γ B6 Δ IC does not form an ordered zipper at adhesion interfaces? If so, is the zipper assembly behavior of γ B4 a unique feature of that isoform, or can this behavior be generalized to other Pcdh γ isoforms? Or, on the contrary, is γ B6 different from all other Pcdh γ isoforms? A sequence alignment across all Pcdh γ genes (perhaps testing the conservation of E497) would help contextualize this important observation. Further, given that Pcdh undergo cis-heterodimerization (Goodman 2022, Thu 2014), can the authors speculate how isoform-specific assembly behaviors impact the organization of lattice comprised of several isoforms?

A sequence alignment of Pcdh γ isoforms around E497 is included in Fig. S6A, which shows that γ B4 is the only isoform that has negative charge (E) at this position, while other isoforms are positively charged (R or K) in γ B subfamily, suggesting that

γ B4 might be unique in forming this zigzag pattern (p.15). However, this does not mean that other isoforms (including γ B6) could not form ordered assembly patterns. Actually according to our unpublished data, different isoforms may form different patterns, which might be relevant to the specific recognition between cells. Similarly, *cis*-heterodimerization (or hetero-oligomerization) of Pcdhs may further expand the repertoire of the assembly patterns on the cell surface, thereby providing structural basis for the specific recognition between cells, as we proposed in the discussion (p.16)

2. Figure 6 would benefit from the inclusion of schematics illustrating the differences between the transfected isoforms.

The schematic illustrations of the transfected isoforms are included in the revised Fig. 6.

3. The authors present data that the VIC and CIC have varying effects on adhesion assemblies, namely, that the VIC exerts a larger impact on the assembly structure. As VIC-dependent trafficking has been observed before in other isoforms (such as Pcdh γ A3) we wonder whether there may be conserved residues within the VIC that may regulate ectodomain assembly across the Pcdh γ subcluster?

To further explore the role of VIC in adhesion, we divided the VIC of γ B4 into three regions: VIC1, VIC2 and VIC3, and each of them was deleted for adhesion assays. The results showed that among the three deletion mutants, γ B4- Δ VIC2 increased the adhesion formation significantly, suggesting that this region in VIC may be important for regulating the assembly of γ B4 and the conserved residues in this region might be important for the assembly of cPcdh γ , which is consistent with the published results showing that residues in this region could affect trafficking and the interaction of the intracellular domains (O'Leary et al., 2011; Ptashnik et al., 2023; Shonubi et al., 2015). These new data are included in the revised manuscript (p. 13 and Fig. S6).

4. The fluorescence colocalization data presented in Figure 6 would be more compelling if accompanied by immunoprecipitation experiments.

In previous publications, the interactions of intracellular domains of cPcdhs have been characterized by biochemical assays including immunoprecipitation. For example: the association of intracellular domain between α 4 and γ A12 or γ B2 (Murata et al., 2004); the association of intracellular domain between γ A3 and γ A3, γ B2, α 1 or β 16 (Shonubi et al., 2015). Based on these results and the conservation of the sequences, interactions of intracellular domains are supposed to be a general feature for cPcdhs, which is consistent with our fluorescence microscopy and EM data.

5. We found lines 250-254 to be confusing and therefore suggest that the authors rewrite this sentence to improve clarity. Further, regarding this data, could the authors quantify the levels of intermembrane densities as Figure S4 is confusing? Maybe the authors can integrate the area under the curve to quantify densities?

The sentence is modified (p.12). The dark densities in the intermembrane space essentially correspond to the cPcdh molecules between the membranes, which can be docked by the crystal structure of γ B4 ectodomain and the docking models provide a better visualization of the molecules at the interfaces (Fig. 1C, 2C-F, 7B, 7D). The histograms of the intermembrane tomographic densities in Fig. S4 also indicate the different patterns of the assembly of γ B4 for the wt and the mutants.

6. I believe that the Bonn. S et paper cited is the one published in 2007, not 2023.

Thanks, the paper is corrected (p.12).

Reviewer #2 (Remarks to the Author):

In the presented manuscript, Zhang, Chen, and Zhang et al. study the structure of the gamma-B4 clustered protocadherin by electron microscopy of plastic-embedded HEK293 cells and complement these studies with immunofluorescence microscopy and mutagenesis studies. By electron tomography, the authors show that the gamma-B4 deletion mutant of the intracellular domain forms ordered clusters in transfected HEK293 cells. These structures are absent when transfecting the full length gamma-B4 constructs. By mutagenesis and subsequent transmission electron microscopy, the authors show that the EC5 domain of gamma-B4 is driving the cis-interactions within the clustered arrays and swapping the domain with the EC5 domain of another protocadherins (gamma-B6) disturbs the interaction and, consequently, the formation of clustered arrays. They demonstrate that a number of point mutation within the EC5 domain, E497K, L585A, and V590G, interferes with array formation, identifying the likely interaction interface of gamma-B4 that leads to cluster formation in absence of the IC domain.

I think it is the presented work is interesting regarding the mediation and regulation of cell-cell interactions. As my expertise is not in the biology of Protocadherins, I will focus myself on the technical aspects of the manuscript while adding comments on the biology and modeling of the observed data wherever the text remains unclear.

Major comments

1. A major concern to me is the model derived in Fig. 2, especially regarding the arrangement of the array in Z. How do the authors justify the incorporation of multiple copies in Z? When going into the cross section of the tomogram, one can

see that the pattern seems to remain the same as in the direction shown in the Figure 2C rather than an array of structures as shown in Figure 2G 90° and 270 °. See an image added for clarification (Image.png in attachment). The current data presented is from ~50-70 nm sections. It would be helpful to perform tomography experiments from thicker sections to obtain more information in the Z direction to clarify the 3D organization of the clustered arrays.

We agree that the arrangement in Z-direction is important for the model. According to the serial sections in Z-direction of an interface on EM specimen, the zigzag pattern remains almost identical, suggesting that the array is maintained in Z. An example of two continuous sections of a zigzag interface is shown in Fig. S5. Actually the thickness of the sections used for tomography is 100 nm (see Material and methods), and we also verify the zigzag molecular model by comparing with the tilts series with different specimen cutting angles (Fig. S3)(p.8).

As for the cross section of the tomogram, since the resolution is relatively low, especially in Z-direction, the feature is not quite clear. Actually the open angle in the image.png is larger than 90°, not the angle of 80° in Fig. 2C or 2F (BTW, the dark “cross-like” (“+”) density is not so close to the membrane and may matches the 270° or 300° views better (Fig. 2G); In the 0° view, the “cross-like” density locates very close to the membrane. Anyways, the feature of cross section is kind of blurred). In addition, the Z arrangement in the tomographic model is similar to that in the crystal packing of the ectodomain, suggesting that such array arrangement is feasible.

2. The authors state that l. 146 - l. 148 "... we also found that the patterns could vary for different interfaces, which might be due to the different cutting angles during EM sectioning as mentioned above." How does this fit to their model? Maybe I missed them, but showing tomograms of these different angles estimating the cutting angle and discussing this more clearly could improve the justification of their model.

In Fig. 2B, we shows an example of an interface visualized at different angles relative to the cutting interface (0°). Three tilt series with different specimen cutting angles are shown in Fig. S3, which shows that the zigzag model match these data reasonably well.

3. It is unclear to me whether the preparations are done correlatively. If so, it would be helpful to show overlays of the clustered arrays with the GFP signal in the supplementary information. If the experiments were not done by CLEM, the authors should comment on how they are certain that the clustered regions are not simply outside the imaged sections and may still be found in earlier or later sections of the same cell.

CLEM can be done quite well for the interfaces during sample preparation if using chemical fixation, but the image resolution is relatively low. The successful rate of CLEM using HPF is lower, therefore the interfaces are tracked by serial sectioning

and checking the assembly patterns under EM by rotating tilt angles. An example of two continuous sections of an interface is shown in Fig. S5.

4. What do the authors suggest is the mechanism of the inhibition of array formation of the intracellular domain of gamma-B4? While it may be beyond the scope of this study, discussing potential mechanisms would be interesting.

We agree that the mechanism of the inhibition of array formation of the intracellular domain is important. According to our EM and LM data and biochemical results published before, the intracellular domain of γ B4 may form homophilic interactions, especially when locating on the cell membrane, which would affect the arrangement of the ectodomain of γ B4 and regulate the assembly patterns at the cell-cell contacts. This is discussed in the manuscript (p.15) and Fig. 8B-C also provide a visualization of the model/mechanism.

5. Which GFP construct was used? Is the construct prone to dimerization and could this have an effect on the observed clustering? Is the effect also observed in non-GFP tagged constructs?

The GFP construct we used is EGFP (enhanced GFP, GenBank: AAB02572) (p.17). We have been using this construct for a long time in different situations and don't see the dimerization caused by GFP. The non-tagged cPcdh has the similar effect for the interfaces.

6. Figure 1 C, Figure 2 G: docking crystal structures into these densities without proper sampling methods is not very helpful. At the minimum, systematic fitting (template matching) should be performed. More rigorous sampling methods would be helpful to get statistics (e.g. with the integrative modeling platform (IMP), Sali Lab) on these assemblies.

The resolution of the tomograms of the specimens prepared by HPF-FS was about ~2 nm, as estimated by the previous publications (McEwen and Marko, 1999). Actually we have tried different docking programs such as UCSF chimera, SITUS, etc., which are commonly used for docking crystal structures into low resolution EM maps (Kawabata, 2018), for model building. The final model was obtained by docking the crystal structure of γ B4_{EC1-6} *trans*-dimer into tomogram densities by UCSF chimera, and the correlation coefficients were 0.71, 0.79, 0.75, 0.76 for γ B4-FL, γ B4- Δ IC, γ B4- Δ CIC and γ B4- Δ VIC, respectively (See Materials and methods, p.19).

7. Regarding the Reporting summary, there is a number of things that need attention. First, in the replication statement, it says "Experiments are reproducible". This is not very informative and should have the number of experiments run per condition.

These information are included in the reporting summary.

8. No "Data availability" statement is given in the manuscript.

We will provide the data upon request (p.20).

Minor comments

1. Figure 1: indicate cytosol and extracellular space of the domain schematic would make the orientation of the domains clearer to the reader.

Ectodomain and intracellular domain are labeled in Fig.1.

2. l.136-137: Figures should be cited in order as it breaks the reading flow, it would be worthwhile to consider rephrasing.

Actually the citations of Fig. 7E and Fig. 8A here are optional, as the information (34 nm for γ B4-FL and 38 nm for crystal packing) are included in the sentences. We tend to cite Fig. 7E and Fig. 8 here because the intermembrane distances of γ B4 constructs were summarized in Fig. 7E and structural models are shown in Fig. 8 (p.7).

3. cPcdh abbreviation introduced in the abstract but not in the introduction

Abbreviation of cPcdh is introduced in the introduction (p.3).

4. Why does the extracellular space get stained by DAPI in the images of Figure 6?

Thanks. We re-prepare the samples for fluorescent microscopy and the new images look fine (Fig. 6).

5. l. 271-273: Sentence structure should be revised.

The sentence is modified (p.13).

6. l. 299-300: These forces (or interactions) are not "not identified" in crystals, but simply not present.

The sentence is modified (p.15).

7. Out of curiosity, how does the structure of cPcdhs compare to the desmosomes studied by the Frangakis group, Sikora et al. <https://www.pnas.org/doi/epdf/10.1073/>

[pnas.2004563117](https://doi.org/10.1073/pnas.2004563117)? *The membrane-to-membrane distance and proposed models seem to be at least similar for the full-length gamma-GB4, thus citing and comparing to these studies may be adequate.*

The intermembrane distances for γ B4 is similar to that of desmosomes, but the *trans* and *cis* interacting modes appear to be different in the two cases, which might be due to their different functional roles in mediating cell adhesion. The paper is cited and discussed (p.15-16).

Reviewer #3 (Remarks to the Author):

This paper represents an important contribution from the He group. Clustered protocadherin ectodomains have been shown to have two dimer interfaces and, together in vitro, they enable the assembly of a zig-zag-like zipper pattern. Whether this zipper pattern exists in vivo is a key question for protocadherin biology. Here, He and collaborators, express clustered protocadherin γ B4 on the surface of HEK 293 cells, and investigate the structures they form by cryo-EM tomography. Protocadherin γ B4 was previously used in crystallographic and liposome-based tomographic studies.

When full-length protocadherin γ B4 protein – containing the cytoplasmic domain – is expressed on the cell surface, and the junctions between apposed cells are examined, no regular zig-zag pattern is observed. The investigators then produced a construct lacking the cytoplasmic domain, and remarkably a zig-zag pattern appears, though it is somewhat different from the crystal-packing lattice previously observed between liposomes with His-tagged protocadherin, having a more obtuse cis-dimer angle. The authors also show that replacement of EC5-EC6, which forms the cis protocadherin dimer interface, leads to the lack of zipper formation.

Based on these observations, the authors propose the quite reasonable suggestion that the cytoplasmic domain can play a role in the structure and assembly of the extracellular region.

The paper is very clearly written, and I have no major scientific issues with the results or their presentation.

Minor issues:

The authors say that EC5-EC6 was “replaced”. Replaced with what?

The sentence is modified (p.9).

Reviewer #4 (Remarks to the Author):

Response letter

Reviewer comments

Reviewer #1 (Remarks to the Author):

The authors have addressed our comments. We therefore recommend this manuscript for publications. Congratulations on an important finding for the field.

Thanks.

Reviewer #2 (Remarks to the Author):

The authors have addressed all my previous comments. The serial section data presented in Figure S5 support the consistency with the suggested model for protocadherin zigzag clustering. I congratulate them on the exciting paper.

Thanks.

Reviewer #4 (Remarks to the Author):

**Tomogram
(as in manuscript)**

Cross Section Model

Cross Section Tomogram

Cross Section Model